# Protective Mechanism Pathway of *Swietenia macrophylla* Extract Nanoparticles against Cardiac Cell Damage in Diabetic Rats

**DOI:** 10.3390/ph16070973

**Published:** 2023-07-07

**Authors:** Rochmah Kurnijasanti, Giftania Wardani, Mohd. Rais Mustafa, Sri Agus Sudjarwo

**Affiliations:** 1Department of Basic Veterinary Medicine, Faculty of Veterinary Medicine, Airlangga University, Surabaya 60115, Indonesia; santisam19@yahoo.com; 2Program Study of Pharmacy, Faculty of Medicine, Hang Tuah University, Surabaya 60239, Indonesia; giftania.wardani@hangtuah.ac.id; 3Department of Pharmacology, Faculty of Medicine, Malaya University, Kuala Lumpur 50603, Malaysia; rais@um.edu.my

**Keywords:** *Swietenia macrophylla*, nanoparticles, antioxidant, cardioprotector, diabetes

## Abstract

Hyperglycemia causes cardiac cell damage through increasing ROS production during diabetic complications. The current study proves the antioxidant activity of *Swietenia macrophylla* (*S. macrophylla*) extract nanoparticles as a protector against streptozotocin (STZ)-induced cardiac cell damage. In this research, high-energy ball milling is used to create *S. macrophylla* extract nanoparticles. The active chemical compounds in the *S. macrophylla* extract nanoparticles were analyzed through phytochemical screening and GC-MS. Furthermore, we characterized the size of *S. macrophylla* extract nanoparticles with Dynamic Light Scattering (DLS). Forty male rats were divided randomly into five groups. In the control group, rats received aqua dest orally; in the diabetic group, rats were injected intraperitoneally with STZ; in the *S. macrophylla* group, rats were injected with STZ and orally given *S. macrophylla* extract nanoparticles. The results of phytochemical screening showed that *S. macrophylla* extract nanoparticles contain saponins, flavonoids, alkaloids, phenolics and tannins. Seven chemical compounds in *S. macrophylla* extract nanoparticles were identified using GC-MS, including phenol, piperidine, imidazole, hexadecene, heptadecanol, dihexylsulfide and heptanol. DLS showed that the *S. macrophylla* extract nanoparticles’ size was 91.50 ± 23.06 nm. Injection with STZ significantly increased malondialdehyde (MDA) levels in cardiac tissue and creatine kinase–myocardial band (CK-MB) and lactate dehydrogenase (LDH) levels in serum. STZ also significantly reduced the expression of nuclear factor erythroid 2-related factor 2 (Nrf2) and the level of superoxide dismutase (SOD) and glutathione peroxidase (GPx) in cardiac tissue compared with the control group (*p* < 0.05). In contrast, the administration of *S. macrophylla* extract nanoparticles can prevent STZ-induced cardiac cell damage through decreasing the level of CK-MB and LDH in serum and the level of MDA in cardiac tissue. *S. macrophylla* extract nanoparticles also significantly increased Nrf2 expression as well as SOD and GPx levels in cardiac tissue. These effects are related to the prevention of cardiac histopathological alteration (degeneration and necrosis) in diabetic rats. These results suggest that *S. macrophylla* nanoparticles contain active compounds such as flavonoids, phenols, piperidine, imidazole and hexadecene and have strong antioxidant activity. These can act as a potential cardioprotective agent against STZ-induced cardiac cell damage due to its antioxidant properties.

## 1. Introduction

Diabetes mellitus is a metabolic disease characterized by hyperglycemia, which is due to the disordered action of insulin on the target organ or its secretion from pancreatic β-cells. Hyperglycemia induces oxidative stress, which contributes to diabetic complications such as retinopathy, nephropathy, neuropathy, atherosclerosis, stroke and cardiac cell damage [1,2].

The condition of oxidative stress is the result of the overproduction of reactive oxygen species/ROS (O_2_^−^, OH^−^ and H_2_O_2_) and a decrease in antioxidant enzyme (SOD, GPx, Catalase) formation, which has an important role in diabetes complications [3,4]. ROS are chemical molecules containing one or more unpaired electron(s) that can interfere with the normal signaling process. These ROS cause cellular damage through their unpaired electron via triggering the oxidant of the molecule and cellular components such as the cell membrane, protein and DNA, which are increased during diabetic complications such as diabetic cardiopathy. Normal physiological processes will also produce ROS, which have a role in cell signaling and tissue homeostasis. However, excessive production of ROS will oxidize lipids, protein and DNA, which is detrimental to cell components, such that the ROS cause necrosis and apoptosis. Overproduction of ROS will activate the process of lipid peroxidation in polyunsaturated fats (PUFA) of the cell membrane that produce lipid peroxides or MDA. High levels of MDA indicate elevated ROS production, which causes cardiac cell damage [5,6,7]. MDA can be used to assess ROS in diabetes mellitus.

In addition, oxidative stress can inhibit Nrf2 from Keap1 and inactivate the antioxidant response element (ARE). This will further decrease the production of antioxidative enzymes, such as SOD, GPx and Catalase. Nrf2 expression and activity are the primary transcription factors that control the production of endogenic antioxidative enzymes for maintaining cellular redox homeostasis [8,9,10].

The use of STZ in a diabetic rat model causes cardiac cell damage, which can increase the level of CK-MB and LDH in the serum. Therefore, the level of CK-MB and LDH in the serum can be used as a marker of cardiac function disorder [11,12,13]. The administration of STZ can also increase MDA levels and decrease Nrf2 expression as well as SOD and GPx levels.

Previous research has shown the benefits and potential of some medicinal herbs in dealing with antioxidants in both the treatment of diabetes and its complications. Referring to world ethnobotany reports, about 800 medicinal herbs, including *S. macrophylla*, are used as traditional treatments for diabetes, since they are considered to have better efficacy with fewer side effects, and they are affordable. *S. macrophylla* has become of great interest and has been used as a traditional treatment against diabetes and its complications. Moreover, the hypoglycemic and antioxidant activities of *S. macrophylla* are supported by evidence from experimental studies [14,15]. Herbal medicine antioxidants are widely used as an alternative exogenous antioxidant to protect cardiac cells in diabetic rats [16,17,18]. *S. macrophylla* is an herbal medicine that has a strong antioxidant effect and can scavenge ROS, such that it can inhibit oxidative stress. The phytochemical analysis of *S. macrophylla* seeds showed the presence of alkaloids, flavonoids, saponin, tannins and phenolic compounds, which may be the active compounds [19,20]. Furthermore, typical phenols that possess antioxidant activity are mainly phenolic acid, flavonoid and tannins. These are able to neutralize free radicals. *S. macrophylla* also has several pharmacological effects, including anti-inflammatory, antioxidant, antiviral, antifungal, antibacterial, immunomodulatory and anti-diabetes properties [21,22,23].

Common problems in natural product antioxidants include bioavailability, solubility, absorption and distribution. To overcome this problem, the development of nanotechnology has led to nanoparticle-sized dosages of natural product antioxidants. Nanobiotechnology is a technology in which substance particles are created on a nanoscale of 10–1000 nm [24,25,26,27]. The use of nanotechnology is expected to increase the therapeutic effects and reduce the toxicity of natural product antioxidants. Considering the anti-diabetes properties and antioxidant activities of *S. macrophylla*, in the current study, we attempt to prove the antioxidant activity of *S. macrophylla* extract nanoparticles as a protector against STZ-induced cardiac cell damage.

## 2. Results

### 2.1. Qualitative Phytochemicals Analysis of S. macrophylla Extract Nanoparticles

The qualitative phytochemical testing of bioactive compounds for the *S. macrophylla* Nanoparticles were presented in Table 1. The finding showed that the phytochemical compounds of *S. macrophylla* extract nanoparticles contained saponin, flavonoids, alkaloids, phenolics and tannins. The data reveal that strong positive results were found for alkaloids, flavonoids and phenolics.

### 2.2. Quantitative Phytochemical Analysis of S. macrophylla Extract Nanoparticles

Determination of the quantity of total phenols, flavonoids and alkaloids of *S. macrophylla* extract nanoparticles has been undertaken as per the methods reported in the literature. The results showed that the *S. macrophylla* extract nanoparticles had a total phenolic content of 59.23 ± 2.41 mg GAE/g extract, flavonoid content of 41.75 ± 3.42 mg QE/extract and alkaloid content of 22.61 ± 1.97 mg CoE/g extract, respectively.

### 2.3. GC-MS Analysis of Bioactive Compounds in S. macrophylla Extract Nanoparticles

GC-MS analysis was performed for bioactive compound profiling in *S. macrophylla* extract nanoparticles, and the results are presented in Figure 1 and Table 2. Seven compounds were characterized in the *S. macrophylla* extract nanoparticles via GC-MS, namely, 1-Heptanol,4-methyl, Dihexylsulfide, Phenol,2,4-bis(1,1-dimethyl), imidazole-4,5-d2, Piperidine, 7-Hexadecene and 1-Heptadecanol, respectively.

### 2.4. The Size Distribution of S. macrophylla Extract Nanoparticles

This research was designed to investigate the protective mechanism pathway of *S. macrophylla* extract nanoparticles against STZ-induced cardiac tissue damage in rats. Characterization via DLS showed that the distribution of the *S. macrophylla* extract nanoparticle size is 91.50 ± 23.06 nm, as seen in Figure 2.

### 2.5. S. macrophylla Extract Nanoparticles’ Effect on Level of CK-MB and LDH in Serum of Diabetic Rats

CK-MB and LDH levels in serum can be used to identify dysfunction and cardiac cell damage. *S. microphylla* nanoparticles’ effect on the level of CK-MB and LDH in the serum of diabetic rats is shown in Table 3. Intraperitoneal injection of STZ significantly increased the level of CK-MB and LDH in serum when compared with the control rats (*p* < 0.05). However, the administration of *S. macrophylla* extract nanoparticles dose-dependently reduced the level of CK-MB and LDH in serum, and only at a dose of 300 mg/kg BW could the decrease be significantly compared with the diabetic group (*p* < 0.05). These results suggest that *S. macrophylla* extract nanoparticles prevent dysfunction and cardiac cell injury in diabetic rats.

### 2.6. S. macrophylla Extract Nanoparticles’ Effect on Cardiac Tissue MDA Levels in Diabetic Rats

The level of oxidative stress can be measured through assessing the production of malondialdehyde (MDA) levels as an indicator of cardiac damage due to oxidative stress, which was caused by increased ROS production. Table 4 shows the efficacy of *S. macrophylla* nanoparticles on MDA levels in cardiac tissue. Rats that were injected with streptozotocin exhibited significantly increased cardiac MDA levels compared with control rats (*p* < 0.05), whereas administration of *S. microphylla* nanoparticles significantly reduced MDA levels in cardiac tissue in a dose-dependent manner.

### 2.7. S. macrophylla Extract Nanoparticles’ Effect on Cardiac Tissue Nrf2 Expression in Diabetic Rats

Nrf2 induces the expression of antioxidants as well as cytoprotective genes, which provoke an anti-inflammatory expression profile, and it is crucial for the initiation of cellular protection against oxidants. The expression of Nrf2 in cardiac tissue was evaluated via immunohistochemistry and can be seen in Figure 3. The gene expression of Nrf2 in the diabetic group was significantly reduced compared to the control group (*p* < 0.05). Administration of *S. macrophyla* extract nanoparticles dose-dependently increases expression of Nrf2 in cardiac tissue, and only at a dose of 300 mg/kg BW can significant increases in Nrf2 expression be compared to the diabetic groups.

### 2.8. S. macrophylla Extract Nanoparticles’ Effect on Cardiac Tissue SOD and GPx Levels in Diabetic Rats

SOD is a first-line antioxidant that catalyzes the dismutation of superoxide anions (O_2_) to hydrogen peroxide (H_2_O_2_), which, in turn, is reduced to oxygen and water by GPx. The level of SOD and GPx in heart tissue can be seen in Table 5. Rats that were injected intraperitoneally with STZ could reduce levels of SOD and GPx in the tissue of the heart significantly when compared with control rats (*p* < 0.05). Meanwhile, pretreatment with *S. macrophylla* extract nanoparticles increased dose-dependent SOD and GPx levels, but only a dose of 300 mg/kg significantly increased SOD and GPx levels in cardiac tissue when compared to diabetic rats (*p* < 0.05).

### 2.9. S. macrophylla Extract Nanoparticles’ Effect on Structural Change in Diabetic Rats’ Cardiac Tissue

Histopathological observations were used to investigate changes in the cardiac cell structure of diabetic rats, as seen in Figure 4. Upon examination with light microscopy, the control rats showed that the cardiac cell structure was normal, while the rats that were given STZ showed irregular morphology, i.e., necrosis of cardiac cells. Administration of *S. macrophylla* extract nanoparticles can protect normal structures and inhibit necrosis of cardiac cells.

## 3. Discussion

Hyperglycemia is a sign of DM that can increase ROS production and can accelerate cardiac cell damage in diabetics [28,29,30]. This study aims to prove the anti-ROS effect of *S. macrophylla* nanoparticles in protecting cardiac cell damage in STZ-induced rats. Many studies have used STZ to model diabetes cardiomyopathy in rats [15,16,17]. In this diabetic cardiomyopathy rat model, elevated ROS levels are presented with higher MDA levels and decreased antioxidant—such as SOD, GPx and Nrf2—expression, which then induced cardiac cell damage [1,5,7].

Our results in this research showed that injection with STZ intraperitoneally can increase the level of MDA and decrease the level of SOD, GPx and Nrf2 expression in cardiac tissue significantly when compared with control rats. An increase in MDA indicates an increase in ROS production. Increased ROS in diabetes will oxidize lipids, proteins and DNA, which can cause damage to cell membranes, disruption of protein function and DNA fragmentation, which results in increased levels of MDA, cell necrosis and apoptosis [2,6,9]. STZ also induced hyperglycemia, which can inhibit antioxidant activity via the inhibition of scavenging, the interaction of glucose with protein, the formation of AGE and blocking receptors, resulting in oxidative cell injury. Streptozotocin can affect the release of insulin from beta cells of islets of Langerhans, which can decrease insulin levels, increase blood glucose and induce diabetic complications such as cardiomyopathy. Another study showed that oxidative stress due to STZ can inhibit Nrf2 from Keap1 and inactivates the antioxidant response element (ARE). This can decrease the production of antioxidant enzymes such as SOD, GPx and Catalase. Nfr2 is an essential transcription factor that controls the response of antioxidants for maintenance in homeostasis on cellular redox [5,12,13].

Nanobiotechnology can be utilized to increase the solubility, absorption, distribution, bioavailability and effectiveness and reduce the toxicity of antioxidant materials [25,26,27]. To prepare *S. macrophylla* nanoparticles, a grinding process was carried out using the ball milling method. This result of the research shows that the manufacture of *S. macrophylla* extract has a nano-size of 91.50 ± 23.06 nm.

Our results indicate that pretreatment with *S. macrophylla* extract nanoparticles only at a dose of 300 mg/kg BW significantly decreases MDA levels, and increases SOD and GPx levels and Nrf2 expression in the cardiac tissue of diabetic rats compared with the diabetic rat group. These results suggest that dose-dependent administration of *S. macrophylla* extract nanoparticles can balance endogenous antioxidants and oxidants so that they can inhibit oxidative stress. Phytochemical screening and GC-MS showed that *S. macrophylla* extract nanoparticles contain active compounds such as flavonoid, phenolic, piperidine, imidazole and hexadecene which have strong antioxidant activity, which suppresses oxidative stress through combating ROS as well as maintaining redox homeostasis.

Increasing the formation of antioxidants and reducing the production of reactive oxygen species (ROS), which can prevent oxidation in polyunsaturated fatty acids on cell membranes and then can decrease MDA levels which can be used as a marker of cardiac tissue damage. This can also be achieved through the activation of Nrf2 by *S. macrophylla* via increasing the antioxidants response element (ARE), which can stimulate the gene transcription that codes endogenic antioxidants enzymes; further, SOD and GPx levels increase.

It has been reported that in vivo and in vitro research shows that *S. macrophylla* scavenges ROS, so it can prevent lipid oxidation, further reducing MDA levels and elevating SOD and GPx levels resulting in a protective effect against oxidative damage in cells [21,22,23]. The administration of natural product antioxidants has been proven to decrease oxidative stress. A recent study shows that Nrf2 has a crucial role in the protection of cardiac cell damage and death caused by oxidative stress in diabetic complications [5,6,7,9].

Intraperitoneal injection with STZ in rats can significantly increment CK-MB and LDH levels compared with control rats. Increasing serum CK-MB and LDH levels can be used as indicators of impaired cardiac function and cell damage, whereas dose-dependent administration of *S. macrophylla* extract nanoparticles significantly decreased levels of serum CK-MB and LDH in diabetic rats. This result shows that pretreatment with *S. macrophylla* extract nanoparticles as an antioxidant prevents cardiac cell damage in diabetic rats. ROS in higher levels in diabetes accelerates cardiac injury, which induces an increase in CK-MB and LDH in serum. Oxidative stress in diabetes has an important role in the progress of cardiac cell damage, which is associated with increased CK-MB and LDH levels. In the same result, exogenous antioxidants can reduce ROS production and can prevent cardiac cell damage through reducing serum CK-MB and LDH levels [1,2,12].

Histological observations clearly showed that the presence of necrosis of rat cardiac cells induced by STZ, conversely administration of *S. macrophylla* extract nanoparticles had a cardioprotective effect through inhibiting necrosis of cardiac cells through antioxidant activity. The same results were also shown by several researchers: administration of STZ can cause necrosis in cardiac cells, and administration of exogenous antioxidants can prevent necrosis of cardiac cells due to STZ. Therefore, *S. macrophylla* extract nanoparticles, which have strong antioxidant effects, are expected to be utilized as protection against diabetes complications, one of which is cardiomyopathy.

## 4. Materials and Methods

### 4.1. Preparation of S. macrophylla Extract

The *S. macrophylla* seed was collected from the Purwodadi Botanical Garden, Indonesia, and identified by a botanist in The Program Study of Pharmacy, Faculty of Medicine, Hang Tuah University, Surabaya, Indonesia. Dried *S. macrophylla* leaves were powdered using a blender. Next, 500 g of powdered leaves was macerated with ethanol at 96%, 2 L for 3 days, and then filtered through a Whatman filter. The filtrate was collected and concentrated in a rotary evaporator at 50 °C. The concentrated extract was dried under open air and stored under refrigeration until further use.

### 4.2. The Manufacturing of S. macrophylla Extract Nanoparticles

The high-energy ball milling method was used to make *S. macrophylla* extract nanoparticles according to the instructions of the nanomachine manufacturer. Then, the *S. macrophylla* extract nanoparticles were characterized via dynamic light scattering (Horiba LA 900, Kyoto, Japan).

### 4.3. Qualitative Phytochemical Screening of S. macrophylla Extract Nanoparticles

The phytochemicals contained in *S. macrophylla* extract nanoparticles can be investigated qualitatively using standard phytochemical screening procedures. Discoloration or the presence of foam can be used as an indicator of the presence or absence of certain phytochemical compounds.

#### 4.3.1. Test for Alkaloids

Two grams of *S. macrophylla* extract nanoparticles was added to 10 mL of 0.1 M hydrochloric acid, warmed in a water bath (50 °C) for 5 min, and filtered trough Whatman filter paper No. 1. After cooling, 3 drops of Dragendorff’s reagent were added and mixed. The appearance of a reddish-brown color is a positive indication of the presence of alkaloids in the sample.

#### 4.3.2. Test for Flavonoids

Two milliliters of *S. macrophylla* extract nanoparticles and five drops of concentrated hydrochloric acid were added. The formation of a red color indicates the presence of flavonoids.

#### 4.3.3. Test for Phenols

*S. macrophylla* extract nanoparticles (0.5 g) were boiled in 5 mL of 70% ethanol in a water bath for 5 min and then filtered through Whatman filter paper No. 1. After cooling, 5 drops of 5% ferric chloride were added and mixed. The appearance of a green precipitate indicates the presence of phenol in the sample.

#### 4.3.4. Test for Saponin

A total of 2 g of *S. macrophylla* extract nanoparticles was dissolved in 5 mL of distilled water. Thereafter, aliquots of 2 mL were taken from *S. macrophylla* extract nanoparticle solution, stirred for 30 s, and briskly agitated. The setups were allowed to settle for 15 min. The presence of frothing, which persists for over 15 min, is an indication of the presence of saponin in the tested sample.

#### 4.3.5. Test for Terpenoids

A quantity of 100 mg of *S. macrophylla* extract nanoparticles was dissolved in 10 mL water. Furthermore, 2 mL of the *S. macrophylla* was taken and then added with 3 drops of concentrated HCl and 1 drop of concentrated H_2_SO_4_. A positive result is indicated by the formation of a red or purple color.

#### 4.3.6. Test for Tanin

For this test, 40 mg of the *S. macrophylla* extract nanoparticles was dissolved with 4 mL water; then, 2 mL was taken, and then we added 1 mL of 10% FeCl_3_. A positive reaction is indicated by the formation of a dark blue or greenish black color.

### 4.4. Quantitative Phytochemical Screening of S. macrophylla Extract Nanoparticles

Quantitative estimations of phenol, alkaloids, flavonoids and tannin contents in the *S. macrophylla* extract nanoparticles were analyzed using methods reported in the literature.

#### 4.4.1. Total Phenols

Total phenols were investigated using the Folin–Ciocalteu method [31]. Approximately 0.5 mL of an ethanol solution of the *S. macrophylla* extract nanoparticles (0.25 mg/mL) was mixed and incubated for 2 min with 2.5 mL of Folin–Ciocalteu reagent (10 times dilution). Furthermore, 2 mL of 7.5% aqueous sodium carbonate (Na_2_CO_3_) was added to the solution, and the mixture was allowed to stand for 30 min at room temperature. The absorbance of the sample was read at 765 nm, and the results were expressed as gallic acid equivalent (mg GAE/g based on dry extract weight).

#### 4.4.2. Total Flavonoids

The AlCl_3_ method is used to determine the total flavonoids [32]. Specifically, 2 mL of the *S. macrophylla* extract nanoparticles at 1 mg/mL concentration was added to 2% AlCl_3_·6H_2_O solution and stood after 1 h incubation at 20 °C. After that, the absorbance was read at 415 nm, and the results were expressed in quercetin equivalent (mg QE/g extract).

#### 4.4.3. Total Alkaloids

Here, 5 mL of pH4.7 phosphate buffer and 5 mL of BCG (Bromocresolgreen) solution were added to 1 mL of *S. macrophylla* extract nanoparticles. The mixture was then vigorously shaken with chloroform before being collected in a 10 mL volumetric flask and diluted with chloroform. In the same manner, as previously described, a set of colchicine reference standard solutions was prepared. A UV-visble spectrophotometer was used to measure the absorbance of test and standard solution against the reagent blank at 470 nm. The total alkaloid content was measured in milligrams of colchicine equivalent per gram (mg CoE/g).

### 4.5. GC-MS Analysis of Bioactive Compounds in S. macrophylla Extract Nanoparticles

*S. macrophylla* solution of 1 μL was injected into GC-MS-QP2010SE, which had a capillary column with a length of 30 mm, a diameter of 0.25 mm, and a thickness of 0.25 μm. Helium carrier gas was added at a flow rate of 1 mL/min with a split ratio 1:50. The pre-programmed oven temperature was 150 °C, and we stored the isothermal for five minutes; the rate of increase was 10 °C/min, and the temperature was increased to 250 °C for five minutes. Compound identification of the GC-MS mass spectrum was performed using the National Institute Standard and Technology (NIST) database. The spectrum components were compared to the NIST data library. The identification of chemical compounds was confirmed based on the peak area and retention time.

### 4.6. Experimental of Animal

Wistar rats with body weights 200–250 g were purchased from LPPT, Universitas Gadjah Mada Indonesia. Rats were kept in plastic cages on a 12 h day/night cycle at a temperature of 26 ± 2 °C and acclimatized for one week before the research. All rats were given food and water ad libitum.

### 4.7. Model of Diabetic Rat

Diabetes was induced via a single injection of STZ intraperitoneally at a dose of 55 mg/kg BW which was dissolved in 0.1 M citrate buffer (pH 4.5). After 3 days of STZ injection, all rats were checked for blood glucose levels with an Accu-check glucometer (Roche Diagnostic). Rats with blood glucose level > 200 mg/dL were considered as diabetic.

### 4.8. Experimental Design

The study utilized rats randomly divided into five groups with eight rats in each group: The control group (rats were given aqua dest); STZ group (rats were injected with a single dose intraperitoneally STZ 55 mg/kg BW); *S. macrophylla* group (Rats were injected intraperitoneally with a single dose of STZ at 55 mg/kg BW, and then after 3 days, rats were given *S. macrophylla* in a dose of 75, 150, 300 mg/kg BW, respectively for 72 days). Rats were euthanized on day 75 with an intraperitoneal injection of ketamine (60 mg/kg) and xylazine (7.5 mg/kg BW). Then, the heart was taken to investigate MDA levels as well as SOD and GPx expression. Histopathological examination of the heart was also performed via Hematoxylin Eosin staining.

### 4.9. Biochemical Estimation of Serum CM-KB and LDH

Serum CM-KB and LDH levels were measured using commercially available test kits (Sigma-Aldrich Co., St. Louis, MO, USA) according to the manufacturer’s instructions

### 4.10. Assessment of Cardiac Tissue MDA Levels

The thiobarbituric acid (TBA) method is utilized to measure MDA in cardiac tissues, which can assess MDA formation using a TBARS assay kit (Company of Cayman Chemical, Ann Arbor, MI, USA). The MDA-TBA complex coefficient was measured with absorbance at 532 nm with the reader of the microplate for assessing MDA levels. The MDA level is expressed in nm/mg tissue.

### 4.11. Immunohistochemical Staining of Nrf2 Expression in Cardiac Tissue

Heart sections (4 μm in thickness) were incubated in 3% H_2_O_2_ for 15 min at room temperature to inhibit endogenous peroxidase activity. Furthermore, the sections were inhibited with normal goat serum for 1 h and then incubated overnight at 4 °C with rabbit polyclonal antibodies specific for Nrf2 (Santa Cruz Biotechnology, Dallas, TX, USA), diluted 1:200 in PBS, 0.01 M, pH 7.2. Control sections were incubated with blocking serum alone. After that, we washed them three times with PBS and incubated them with a secondary antibody from the Ultra Vision Quanto Detection system HRP DAB (Therma Fisher Scientific, Waltham, MA, USA) for 30 min at room temperature and with 3 3′ diaminobenzidine (DAB) color reagent. All slides were scored; for each slide, ten microscopic viewing fields were examined at 400× magnification and scored as follows: no immunopositive cells were given a score of 0; immunopositive cells between 1 and 25% were given a score of 1; immunopositive cells between 26 and 50% were given a score of 2; immunopositive cells between 51 and 75% were given a score of 3; sections with more than 75% immunopositive cells were given a score of 4.

### 4.12. Assessment of Cardiac Tissue SOD and GPx Expression

To assess the SOD enzymatic activity in rat cardiac tissue, protein from the cardiac was extracted and assessed according to the procedure of Bradford. SOD inhibition decreased nitro blue tetrazolium (NBT) (Sigma-Aldrich, USA) in each sample, determined via spectrophotometry at 560 nm. SOD levels are shown as U/mg protein.

To assess levels of GPx, the samples were incubated with NaN_3_ and H_2_O_2_. The homogenate 0.1 mL of cardiac tissue was incubated with ethylene diamine tetraacetate 0.2 mL, sodium azide and H_2_O_2_ mixed with phosphate buffer. The mixture was centrifuged at 200 rpm and stopped through adding reagent TCA. The supernatant was mixed with disodium hydrogen phosphate and DTNB, then incubated for 10 min at 37 °C. The absorbance was measured at 412 nm after the color was formed. The levels of GPx are shown as U/mg protein.

### 4.13. Histopathological Observations

At the end of the study, all rat hearts were fixed in buffer formalin 10% and embedded with paraffin. The section of the heart tissue was 4 μm stained with Hematoxylin and Eosin. Histopathological observation of the heart was carried out with a light microscope to determine the presence of kidney cell damage such as degeneration and necrosis.

### 4.14. Statistical Analysis

The results are shown as the mean ± standard error of the mean (SEM) and were analyzed using one-way analysis if variance (ANOVA) followed by the Duncan multiple comparison test using SPSS 21. Differences in means were considered significant at *p* < 0.05.

## 5. Conclusions

*S. macrophylla* extract nanoparticles function as cardioprotection in diabetic rats via an antioxidant effect through inhibiting MDA production and elevating the expression of Nrf2, SOD and GPx in heart tissue. In addition, *S. macrophylla* extract nanoparticles also reduce serum CK-MB and LDH in diabetic rats. *S. macrophylla* extract nanoparticles also contain active compounds such as flavonoid, phenolic, piperidine, imidazole and hexadecene, which have strong antioxidant activity suppress oxidative stress through combating ROS as well as maintaining redox homeostasis. Several researchers reported that the size of the extract nanoparticles is smaller than the extract so that the absorption, biodistribution, specificity, sensitivity and pharmacological activity of the extract nanoparticles are better than the extract. However, in this study, we only had a group of diabetic rats that were given *S. macrophylla* extract nanoparticles, and no group of diabetic rats that were given *S. macrophylla* extract, so we could not distinguish the effectiveness and efficiency between the *S. macrophylla* extract nanoparticles and *S. macrophylla* extract. Therefore, it is necessary to carry out research that proves the differences in effectiveness, efficiency and pharmacological activity between *S. macrophylla* extract nanoparticles and *S. macrophylla* extracts.

## Figures and Tables

**Figure 1 pharmaceuticals-16-00973-f001:**
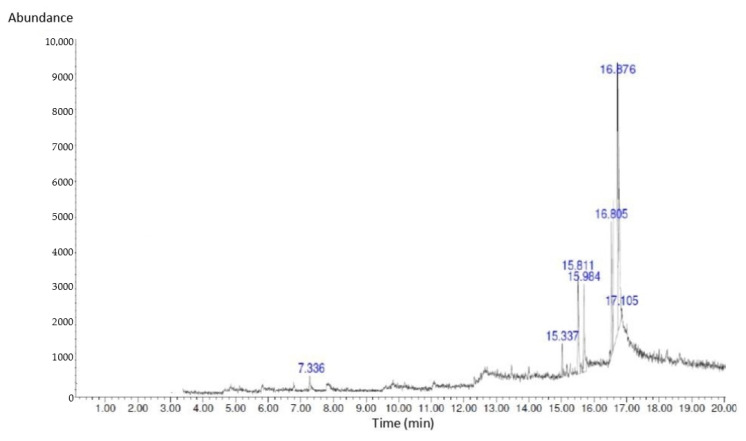
GC-MS chromatogram of *S. macrophylla* extract nanoparticles.

**Figure 2 pharmaceuticals-16-00973-f002:**
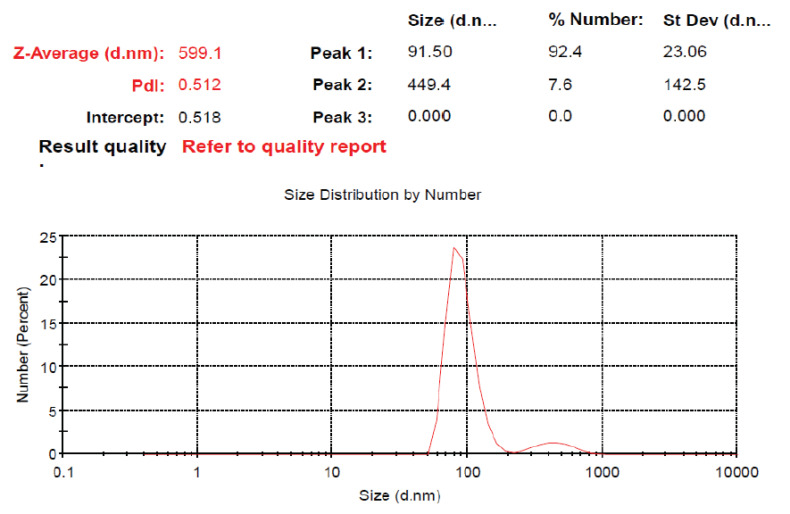
Size distribution of *S. macrophylla* extract nanoparticles.

**Figure 3 pharmaceuticals-16-00973-f003:**
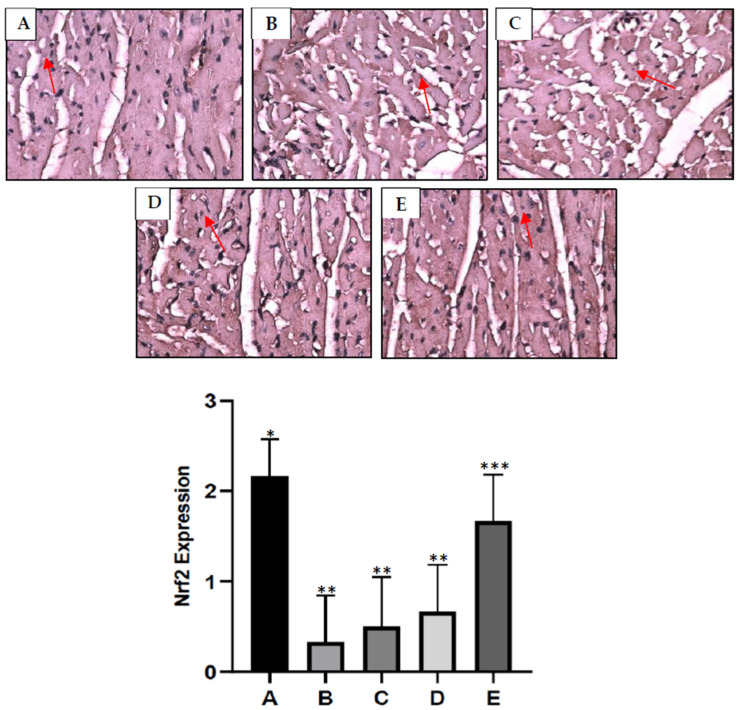
Immunohistochemical of rat cardiac tissue. Nrf2 expression (red arrow). Control rat (**A**); a significant decrease in Nrf2 expression in cardiac tissue was seen in the diabetic rats compared to the control rats (**B**); the administration of *S. macrophylla* extract nanoparticles at a dose of 75 mg/kg BW and 150 mg/kg BW still showed a decrease in Nrf2 expression (**C**,**D**); the administration of *S. macrophylla* extract nanoparticles at a dose 300 mg/kg BW significantly increased Nrf2 expression in cardiac tissue (**E**). *, **, *** The columns with different letters show significance between groups (*p* < 0.05).

**Figure 4 pharmaceuticals-16-00973-f004:**
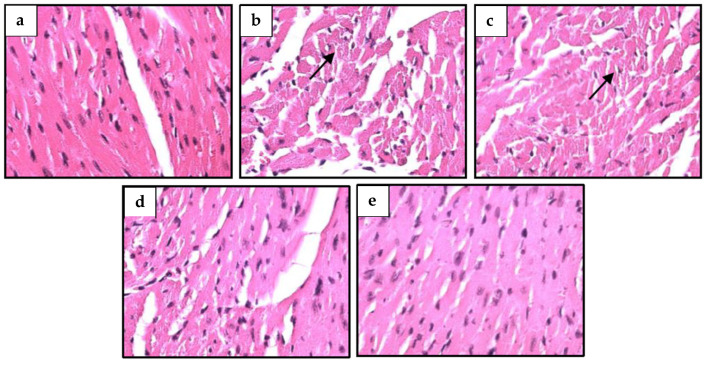
Histological of cardiac tissue of rats. Control rats showed morphology structure of rat cardiac cells is normal (**a**); necrosis (black arrow) was observed in diabetic rats’ hearts (**b**); pretreatment with *S. macrophylla* extract nanoparticles at dose of 75 mg/kg and 150 mg/kg bw in diabetic rats still indicated mild necrosis (**c**,**d**); meanwhile, a dose of 300 mg/kg can prevent necrosis in the diabetic rat heart (**e**). H&E, 400×.

**Table 1 pharmaceuticals-16-00973-t001:** Phytochemicals of bioactive compounds in the *S. macrophylla* extract nanoparticles.

No	Phytochemicals	Presence
1	Phenols	+++
2	Flavonoids	+++
3	Alkaloids	++
4	Saponins	++
5	Terpenoids	++
6	Tannins	++

Intermediate positive: ++; Strong positive: +++.

**Table 2 pharmaceuticals-16-00973-t002:** The GC-MS analysis of *S. macrophylla* extract nanoparticles.

No	Compound Name	RT (min)	Peak (%)	Peak Area
1	1-Heptanol,4-methyl	7.336	2.16	217,421.25
2	Dihexylsulfide	15.337	4.13	512,427.28
3	Phenol,2,4-bis(1,1-dimethyl)	15.811	7.56	839,873.46
4	Piperidine	15.984	10.94	1,345,839.14
5	Imidazole-4,5-d2	16.805	11.75	1,701,547.29
6	7-Hexadecene	16.876	15.64	1,936,913.72
7	1-Heptadecanol	17.105	5.13	613,201.15

**Table 3 pharmaceuticals-16-00973-t003:** Effect of *S. macrophylla* extract nanoparticles on serum CK-MB and LDH levels in diabetic rats.

Group	Mean ± SD	
	CK-MB	LDH
Control Rats	78.4 ^a^ ± 2.53	108.7 ^a^ ± 3.41
Diabetic Rats	107.6 ^b^ ± 2.02	158.8 ^b^ ± 5.56
*S. macrophylla* Nano 75 mg/kg BW	108.7 ^b^ ± 3.04	154.2 ^b^ ± 4.44
*S. macrophylla* Nano 150 mg/kg BW	105.2 ^b^ ± 6.24	151.7 ^b^ ± 2.98
*S. macrophylla* Nano 300 mg/kg BW	91.7 ^c^ ± 2.85	133.7 ^c^ ± 2.99

^a–c^ The different superscript in each column shows significant difference between the mean (*p* < 0.05).

**Table 4 pharmaceuticals-16-00973-t004:** Effect of *S. macrophylla* extract nanoparticles on cardiac tissue of MDA levels in diabetic rats.

Group	Mean ± SD
	MDA (nmol/mg Tissue)
Control Rats	50.8 ^a^ ± 4.02
Diabetic Rats	76.7 ^b^ ± 4.32
*S. macrophylla* Nano 75 mg/kg BW	80.0 ^b^ ± 2.83
*S. macrophylla* Nano 150 mg/kg BW	75.5 ^b^ ± 4.18
*S.macrophylla* Nano 300 mg/kg BW	60.5 ^c^ ± 3.08

^a–c^ The different superscript in each column shown significant difference between the mean (*p* < 0.05).

**Table 5 pharmaceuticals-16-00973-t005:** Effect of *S.macrophylla* extract nanoparticles on cardiac tissue SOD and GPx levels in diabetic rats.

Group	Mean ± SD	
	SOD (U/mg Protein)	GPx (U/mg Protein)
Control Rats	13.83 ^a^ ± 1.60	2.73 ^a^ ± 0.25
Diabetic Rats	6.67 ^b^ ± 0.82	0.78 ^b^ ± 0.08
*S. macrophylla* Nano 75 mg/kg BW	6.33 ^b^ ± 0.81	0.85 ^b^ ± 0.05
*S. macrophylla* Nano 150 mg/kg BW	7.17 ^b^ ± 0.75	0.88 ^b^ ± 0.08
*S. macrophylla* Nano 300 mg/kgBW	9.17 ^c^ ± 0.76	1.68 ^c^ ± 0.31

^a–c^ The different superscript in each column shows significant difference between the mean (*p* < 0.05).

## Data Availability

Data are presented within the article.

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
