# Peer review of "Protective Mechanism Pathway of Swietenia macrophylla Extract Nanoparticles against Cardiac Cell Damage in Diabetic Rats"

_pharmaceuticals, 2023, doi:10.3390/ph16070973_

Round 1
Reviewer 1 Report
The present manuscript report the cardioprotective effects of a nanoformulation from Swietenia macrophylla seeds extract in an experimental model of diabetes. The subject is an interesting issue within the Pharmaceuticals' scope. However, the manuscript has some flaws which raise some questionable issues.
First of all, the reduction of the dosis necessary to achieve a therapeutic effect would be expected with nanotechnology. In this work, it is not clear the effects of the regular extract and the potential advantages of nanoformulation in this particular case. Furthermore, 300 mg/kg is a high dose for an extract, even possible, a relevant result would be achieved it nanotechnology would markedly reduce this doses in order to have an significant effect.
Other point to be elucidated is the chemical composition; GC-MS seems not to be an adequate method to assess this type of material. NIST library does not provide an accurate source of variable phytochemical classes, but only hydrocarbons. Likewise, the phytochemical classes identified in preliminar screening were not resemble to an analysis bu GC-MS. For this issue, a LC-MS analysis is more adequate.
In results section, no statistical difference was indicated in tables or figures, but only in manuscript at some points. Moreover, the significant effects were very slight, besides the use of nanotechnology would be expected to promote a improvement of benefits of the extract of Swietenia macrophylla. This point is not considered in the manuscript, which reports solely the effects without any further consideration.
As minor observation, the method used to assses GPx is suitable to measure GSH (reduced glutathione) instead of GPx. This point must be revised.
In my opinion, this manuscript is not suitable for publication in Pharmaceuticals. As Editor's discretion, the manuscript could be suggested to the sister journal Drugs and Drug Candidates.
Author Response
Respond to comments of reviewer 1
The preparations we use for research are extracts that contain lots of ingredients such as phenol, flavonoids, alkaloids, terpenoid, and tannins, and the active compound content is small, so the dose for extracts is generally large, namely 500 mg/kg BW to 1000 mg/kg. kg BW while the dose of nano extract that we use has a significant effect at a lower dose, namely 300 mg/kg BW
We use extracts in research, therefore we carry out phytochemical screening. However, to find out the profile of the active compound, we used GC-MS. Actually, LC-MS is very good, but in our department, we don't have one
Thank you for your advice, sorry I forgot to add a superscript that can show statistical differences between groups in tables or figures. We have fixed it according to your suggestions
In general, for research related to oxidative stress, the antioxidants assessed are SOD and GPx because these have an important role, not GSH.
If we read the articles published in Pharmaceuticals, there are those that are similar to our articles. But I'm waiting for a decision from the editor

Reviewer 2 Report
The manuscript entitled “Protective mechanism pathway of Swietenia macrophylla extract nanoparticles against cardiac cell damage in diabetic rats” shows interesting information for applications of herbal extract in health promotion. However, there are some comments and questions about this manuscript as follows:
1. What kind of nanoparticles was used for delivery of Swietenia macrophylla extract (SME)?
2. The authors should inform more details of the nanoparticles in the introduction section, in particular, the formulation of the nanoparticles.
3. How was the SME nanoparticles sterilized before being injected to the rats? Please specify in the materials and methods section.
4. The authors should show micrographs of the SME nanoparticles in the manuscripts.
5. Please inform an approval number of the ethic in laboratory animal use.
6. The amount of an extract and reagents shown in the section of materials and methods should be presented in forms of a word of quantity instead of a digit/figures at the beginning of each sentences.
7. Line 280; please specify the solvent (s) used of maceration of Swietenia macrophylla at a concentration of 96%.
8. Please check the chemical formulations of each reagent used in this manuscript. The atom numbers of each element should be presented as a subscript.
Author Response
Respond to comments of reviewer 2
- In this study we made nano extract using a high-energy ball milling method, so we did not use it for extract delivery
- In this study, there was no nano extract formulation because the nano extract was made using the high-energy ball milling method, unlike the preparation of nano using the ionotropic gelation
- Administration of SME nanoparticles is not injected but given orally. The extract is made by maceration with ethanol, therefore the material used is sterile. For the tools we use for research, we sterilize them with an autoclave
- Thank you for your advice and I have added that the size of the SME nanoparticles is 91.50 ± 23.06 nm in the manuscript
- I have added the ethical approval number for the use of laboratory animals.
- I have changed the amount of extracts and reagents shown in the materials and methods section, we present them in the form of the word quantity
- Online 280; I have added the solvent used for maceration of Swietenia macrophylla with ethanol at 96 % concentration
- I have examined the chemical formulation of each reagent used in this manuscript and the atomic number of each element, we present as a subscript.

Reviewer 3 Report
Swietenia macrophylla extract have been known to have the substantial anti-diabetic and antioxidant properties. This study investigated the therapeutical effect of Swietenia macrophylla extract nanoparticles against cardiac cell damage in diabetic rats.
Comments:
1. The Swietenia macrophylla extract was collected and concentrated in a rotary evaporator at 50 oC. Would the bioactivity of those antioxidant components of Swietenia macrophylla?
2. In Figure 3A-E, it is difficult to tell the signal differences by eye.
3. The S. macrophylla extract treatment group should be included to show the delivery efficiency difference between extract and nanoparticles.
Author Response
Respond to comments of the reviewer 3
- The results of the Phytochemical Screening and GC-MS studies showed that the Swietenia macrophylla extract nanoparticles contained Phenol, Flavonoids, piperidine and hexadecene which are based on references have antioxidant effects
- In Figure 3A-E, it is difficult to see the difference in signal with the naked eye, therefore we enlisted the help of a pathologist to check the results of these immunocytochemistry.
- Very good advice, but in the results of this study we did not compare it with extracts. For further research, we will compare extracts and extract nanoparticles

Round 2
Reviewer 1 Report
I agree with the authors about relevance of determination of GPx instead of GSH. The issue raised in my report is based on the experimental procotol using DTNB is widely reported as a protocol to determine GSH, not GPx. Then, if authors have intented to determine GPx, the choice of experimental protocol was not adequated. Therefore, if this protocol was duly used in this work, GSH was determined, not GPx activity. Please check this protocol in https://doi.org/10.1016/0003-9861(59)90090-6.
Author Response
Answer Reviewer 1
Thank you very much for the corrections on my very thorough article, especially on measuring GPx levels. It is true that DTNB is commonly used to measure GSH but GPx measurement is also related to GSH as the method reported below:
- Glutathione peroxidase (GPx) is another antioxidant enzyme that catalyzes the reduction of hydrogen peroxide and lipid peroxides to water and their corresponding lipid alcohols via the oxidation of reduced glutathione (GSH) into glutathione disulfide (GSSG). Its activity can be assessed by the method of Rotruck et al. [137], as described by Hafeman et al., in which the samples are incubated with hydrogen peroxide in the presence of glutathione for a particular time period. The amount of utilized hydrogen peroxide is then determined by directly estimating GSH content using Ellman’s reagent, 5,5-dithiobisnitrobenzoic acid (DTNB). Another method developed by Kokatnur and Jelling and later described by Paglia and Valentine and Pleban et al. relies on a similar principle, with GPx catalyzing the oxidation of glutathione by cumene hydroperoxide (for selenium-independent GPx) or hydrogen peroxide (for selenium-dependent GPx). However, in this method, the oxidized glutathione is later reduced by exogenous glutathione reductase causing the coenzyme of the reaction, NADPH, to become oxidized into NADP+. The change in the absorbance can then be read spectrophometrically at λ = 340 nm. (Oxidative Medicine and Cellular Longevity, Volume 2019, https://doi.org/10.1155/2019/1279250)
- Glutathione peroxidase (GPx) was measured according to the modified Rotruck et al. method. First, we poured, in order, 200 mL of 0.4M tris-HCl (pH 7), 100 mL of 1mM NaN3, 200 mL of the sample, 200 mL of 2mM glutathione, and 100 mL of 0.2mM H2O2 into a tube. The tubes were kept at 37 _C for 10 min, after which we added 0.4 ml of 10% TCA to the tubes. The tubes were centrifuged at 2000 rpm for 3 min. A total of 25 mL of the supernatant was poured into an ELISA microplate where it was mixed with 140 mL of 0.2M tris-EDTA (pH 8) and 30 mL of DTNB. After 30 min incubation at room temperature, an ELISA reader was used to obtain the absorbance of each sample in triplicate at 420 nm versus a blank. The GPx level was reported as U/mg. Sodium citrate was used as the solvent for DTNB (Archives of Physiology and Biochemistry, DOI: 10.1080/13813455.2021.1877308)
Please help me so that my article becomes better and can be published in Pharmaceuticals. Thank you very much for your attention and cooperation

Reviewer 2 Report
The authors responded to the suggestions from reviewers. The manuscript thus could be accepted in the present form.
Author Response
Thank you very much for your help and cooperation. Very kind reviewers with very thorough suggestions so that my article becomes better and can be accepted and published in Pharmaceuticals
